# In Vivo Evaluation of Bone Regenerative Capacity of the Novel Nanobiomaterial: β-Tricalcium Phosphate Polylactic Acid-co-Glycolide (β-TCP/PLLA/PGA) for Use in Maxillofacial Bone Defects

**DOI:** 10.3390/nano14010091

**Published:** 2023-12-28

**Authors:** Mrunalini Ramanathan, Ankhtsetseg Shijirbold, Tatsuo Okui, Hiroto Tatsumi, Tatsuhito Kotani, Yukiho Shimamura, Reon Morioka, Kentaro Ayasaka, Takahiro Kanno

**Affiliations:** Department of Oral and Maxillofacial Surgery, Shimane University Faculty of Medicine, 89-1, Enya-Cho, Izumo 693-8501, Shimane, Japan; m219432@med.shimane-u.ac.jp (M.R.); m239403@med.shimane-u.ac.jp (A.S.); tokui@med.shimane-u.ac.jp (T.O.); tatsumi@med.shimane-u.ac.jp (H.T.); kota0192@med.shimane-u.ac.jp (T.K.); shima06@med.shimane-u.ac.jp (Y.S.); moriokareon@med.shimane-u.ac.jp (R.M.); ayasakakentaro@med.shimane-u.ac.jp (K.A.)

**Keywords:** β-tricalcium phosphate, β-TCP/PLLA/PGA cotton-like fiber, maxillofacial bone regeneration, synthetic/artificial bone graft substitutes, electrospun nanobiomaterial

## Abstract

Maxillofacial bone defects are treated by autografting or filling with synthetic materials in various forms and shapes. Electrospun nanobiomaterials are becoming popular due to their easy placement and handling; combining ideal biomaterials extrapolates better outcomes. We used a novel electrospun cotton-like fiber made from two time-tested bioresorbable materials, β-TCP and PLLA/PGA, to check the feasibility of its application to maxillofacial bone defects through an in vivo rat mandibular bone defect model. Novel β-TCP/PLLA/PGA and pure β-TCP blocks were evaluated for new bone regeneration through assessment of bone volume, inner defect diameter reduction, and bone mineral density. Bioactive/osteoconductivity was checked by scoring the levels of Runt-related transcription factor x, Leptin Receptor, Osteocalcin, and Periostin biomarkers. Bone regeneration in both β-TCP/PLLA/PGA and β-TCP was comparable at initial timepoints. Osteogenic cell accumulation was greater in β-TCP/PLLA/PGA than in β-TCP at initial as well as late phases. Periostin expression was more marked in β-TCP/PLLA/PGA. This study demonstrated comparable results between β-TCP/PLLA/PGA and β-TCP in terms of bone regeneration and bioactivity, even with a small material volume of β-TCP/PLLA/PGA and a decreased percentage of β-TCP. Electrospun β-TCP/PLLA/PGA is an ideal nanobiomaterial for inducing bone regeneration through osteoconductivity and bioresorbability in bony defects of the maxillofacial region.

## 1. Introduction

Leading causes of bone defects in the maxillofacial region include trauma sustained to the facial region and iatrogenic bone resection for extirpation of pathology such as cysts, tumors, and malignant lesions. Systemic causes of bone mass loss include congenital abnormalities, diseases, and medications [1]. The presence of incomplete/irregular bony segments in the maxillofacial region not only leads to esthetic and functional impairments but also invariably affects the patient’s quality of life (QOL) [2]. The requirement to replace autologous bone/bone substitutes in acquired or iatrogenic bone defects is still quite a challenge in reconstructive surgery. Autologous bone grafting remains the gold standard for bridging and filling bone defects to date and has been employed for over a century; approximately 2 million autografts are being placed every year in surgical specialties across the world [3]. Bone grafting, with or without the addition of titanium osteosynthesis, provides the necessary mechanical support while promoting bone regeneration through osteoinduction and osteoconduction [4]. It is associated with a multitude of complications, such as secondary surgery at an unrelated site, donor site morbidity, infection, loss of sensory perception, pain, difficulties with contour adaptation, and the possible need for re-operation at the recipient site due to potential graft failure [4,5,6].

Bone graft substitutes are synthetic/artificial biomaterials specifically engineered to avoid the risks that come along with autologous bone harvesting. The synthetic biomaterial used as an autologous bone graft substitute should allow osteogenic cells to proliferate within them (bioactive/osteoconductivity), be biocompatible with host tissues, and possess biosafety [7,8].

Natural bone is composed of apatite calcium phosphate minerals, and ceramic scaffolds that mimic the natural bone morphology have been used for several decades with good success rates [9]. The most popular calcium phosphate ceramic-based synthetic nanobiomaterials that are employed clinically are subclassified into hydroxyapatite (HA) and β-tricalcium phosphate (β-TCP) [10]. These scaffolds have biosafety, good mechanical properties, ideal degradation time, excellent bioactivity (Mesenchymal Stem Cell—MSC attraction), and a micro-environment that promotes angiogenesis, allowing for bone replacement and tissue growth [11,12]. HA bone substitute shows ideal features in being non-toxic, osteoconductive, having micropores to support tissue ingrowth, and maintaining stability within host tissues. It is, however, discredited because of its brittleness, decreased flexibility, fatigue due to the inability to withstand the dynamic forces of bone [13], and poor resorbable characteristics [14]. Hence, the utilization of HA bone substitute as a sole choice for bone regeneration in maxillofacial defects is limited. β-TCP bone substitute is highly favored owing to its osteoinductive (stimulation of stem cells to proliferate into osteogenic cell lineage) and osteoconductive (bone growth onto/into a suitable surface/material) properties [9]. It has an interconnected porous network that allows cell proliferation, angiogenesis, and eventually bone tissue growth. With β-TCP bone substitute, degradation inside host tissues is smooth, and there is a faster release of calcium ions in the scaffold vicinity [15]. Although the positive aspects of β-TCP bone substitute are superior, it still has a profound morphological limitation in being brittle and lacking plasticity [16].

Biodegradable materials have been in existence since the 1990s [17]. Polyglycolic acid (PGA), followed by poly-l-lactic acid (PLLA) (1st generation), were the first materials to be analyzed for their clinical efficacy [18]. Due to chronic inflammatory reactions and prolonged degradation observed in vivo, 2nd generation bioresorbable materials were developed by combining and adjusting the ratio of PLLA and PGA polymers [19]. The resultant material, polylactic acid-co-glycolide (PLLA/PGA), exhibited a shorter degradation time but inadequate strength during the bone-healing phase and a lack of bioactive/osteoconductivity properties. Thereafter, u-HA was added to 1st generation (PLLA) and 2nd generation (PLLA/PGA) copolymers to induce bioactive/osteoconductivity, making up the 3rd and 4th generation bioresorbable materials, respectively [20]. When used in maxillofacial osteosynthesis, these materials show adequate strength during healing and also promote new bone regeneration, as evidenced by our previous research work [18,21,22,23]. Handling the characteristics of the above-mentioned bioresorbable materials renders easy manipulation and placement inside maxillofacial bone defects [24].

The electrospinning technique gained popularity after the introduction of an apparatus that utilized electric charge to obtain synthetic fibers by Formhals in 1934 [25]. The electrospinning technique generates nanofibers from polymers in the range of 3–5000 nanometers [26,27] with good surface area and causes an increase in the surface roughness of the scaffold. The fiber diameter is inversely proportional to the melting temperature of the polymer [25]. The apparatus generally consists of a needle, a syringe, a voltage source with negative or positive polarity, a collector, and a controlled pump. When the syringe containing a polymeric mixture is exposed to a high voltage, it produces fibers. Compared to other techniques for fiber production (self-assembly, fibrillation, gas-jet, and nanolithography), electrospinning is relatively economical and easy to perform [26]. Setup parameters such as pressure, humidity, temperature, voltage, distance between the syringe and collector, polymer viscosity, and conductivity are important factors to be controlled [26]. Of note is the coaxial spinning technique, which utilizes two nozzles, with a smaller nozzle present on the inside of a larger nozzle that results in the encapsulation of the contents of the smaller nozzle. Coaxial spinning is found to be of more value in drug delivery systems [25]. It is said that electrospun fiber-cross can simulate pore structure similar to that of scaffolds; this helps in nutrient permeation to the newly regenerated tissues [28]. Electrospun fiber mats [29] and membranes have also been fabricated and employed in wound healing [30]. The structure of these fibers can be comparable in their morphology to the extracellular matrix (ECM) in vivo [31], hence allowing easy cell attachment, proliferation, and differentiation. Electrospinning does not necessarily change the inherent property of the synthetic material but rather is a modality to boost the performance of the scaffold [32].

To enable better handling for clinical applications, β-TCP bone substitute has been combined with either PLLA or PLLA/PGA through the electrospinning method, and elegant new nanobiomaterials have been introduced [33,34,35]. These electrospun nanobiomaterials have a cotton-like consistency, can increase in volume and gain stiffness by blood absorption, and permit easy packing [35] into bone defects. Given the morphological complexity of the maxillofacial region, packing of β-TCP/PLLA/PGA fibers is much more convenient, and the nanobiomaterial can be placed inside larger and uneven bone defects, which is not feasible when using plain β-TCP bone substitute because it has a very brittle structure and hence, limited application. β-TCP/PLLA/PGA is more advantageous to β-TCP in consideration of the fact that it is very hydrophilic, anti-bacterial, can be easily deformed, and does not fall out of the defect site after placement [35]. PLLA/PGA has a faster degradation time due to its hydrophilic nature, is completely excreted from the host, and is shown to produce less inflammation than pure PLLA [36], but is devoid of any bioactive properties. The higher molecular weight of PLLA/PGA enables easy production of fibers and allows interspersing of β-TCP particles.

Several in vitro analyses of β-TCP/PLLA/PGA electrospun fibers have been carried out, and the material is reported to be non-cytotoxic, allows ample migration of osteogenic cells, and is said to be an ideal material for guided bone regeneration (GBR) [37,38,39]. Castro et al. [39] showed that electrospun fibers composed of 20% β-TCP and PLLA/PGA demonstrated excellent cytocompatibility with more than 80% metabolic activity. Various concentrations of β-TCP with PLLA/PGA have been tested [40,41], and it is worthy to note that even the lowest concentrations demonstrate excellent biocompatibility. Bioactivity increases with β-TCP addition, with cell metabolic activity rising only a day after implantation. Cell survivability with electrospun fibers is also marked, indicating good material-cell adhesion. Isaji et al. [40] in their in vitro study have remarked that combining β-TCP with PLLA/PGA may decrease the overall tensile strength of the material; the proportion of β-TCP in the fiber is inversely proportional to the tensile strength. The absence of a chemical bond between β-TCP and PLLA/PGA could be attributed to the same. Hydrophilicity is an important requisite of biodegradable materials, and the addition of β-TCP aided in improving the hydrophilicity of the nanobiomaterial, mainly due to its cotton-like structure. β-TCP contributes to the rough surface of the fibers. Coarse fiber structure enables the attachment and proliferation of osteogenic cells. Β-TCP addition increases surface area, which promotes cell attachment and growth. β-TCP/PLLA/PGA nanobiomaterial has so far demonstrated good biocompatibility and is FDA-approved for use in human patients [42]. Researchers have shown that β-TCP/PLLA/PGA scaffolds can be used for extended drug delivery and filling bone defects to promote osteoinductive effects as well as provide a suitable environment for cell migration and proliferation in vivo [42]. Electrospun biomaterials such as β-TCP/PLLA/PGA and β-TCP/PLA have been applied to regeneration of the periodontium [43] and to aid bone regeneration in patients with gingival recession [38]. Electrospun fibers are being studied in orthopedic surgery as well; electrospun β-TCP/PLLA/PGA combined with bone marrow concentrate showed consistent long-term bone formation in a rabbit spinal fusion model [41]. PLLA/PGA and β-TCP have also been studied as drug delivery systems alongside growth factor [44] and stem cell application—all of which have shown favorable results in terms of regenerative therapy [26,45].

Though the properties of β-TCP/PLLA/PGA scaffold and the electrospun cotton-like fiber variant have been tested and studied previously, there is no basic research that compares the effects of the electrospun version of the novel nanobiomaterial, β-TCP/PLLA/PGA, with β-TCP bone substitute for maxillofacial bone defect regeneration in vivo anywhere in the world. We conducted this study for the purpose of defining bone regeneration characteristics and evaluating the osteoinductive and osteoconductive properties of β-TCP/PLLA/PGA against the time-tested standard β-TCP bone substitute in vivo.

## 2. Materials and Methods

### 2.1. Nanobiomaterials Used

We used two different types of nanobiomaterials in this study. The first was β-TCP/PLLA/PGA, or ReBOSSIS-MT (ORTHOReBIRTH, Kanagawa, Japan). The nanobiomaterial had a white cotton shape with electrospun fibers of PLLA/PGA (higher molecular weight) interspersed with β-TCP particles in the ratio of 30 wt% and 70 wt%, respectively. β-TCP particles had an average size of 1–5 μm. The volume of β-TCP/PLLA/PGA used was 3 mg. β-TCP/PLLA/PGA nanobiomaterial was produced in the following manner: The specific concentration of the materials mentioned above was thoroughly kneaded using a machine to provide a homogenous mixture. The paste formed was then diluted with chloroform and fed into an electrospinning machine to obtain fine cotton-like fibers. The second nanobiomaterial was a pure 100% β-TCP bone substitute block (OSferion^®^), kindly provided by Teijin Medical Technologies (Osaka, Japan) and manufactured by Olympus Terumo Biomaterials Corp. (Tokyo, Japan). The block had dimensions of 4 mm in diameter and 2 mm in thickness. The porosity of the block was 77.5 ± 5%, the weight was 0.016 ± 0.001 g, and the density was 0.69 ± 0.06 g/cm^3^. To view the material under a scanning electron microscope (SEM), non-sterilized electrospun fibers were coated with platinum palladium alloy and fixed using carbon tape in a sample stand. Macroscopic and SEM images of the nanobiomaterials are displayed in Figure 1.

### 2.2. Animal Protocol: Creation of Critical Size Mandubular Defect, Packing Nanobiomaterials, and Sacrifice

Institutional review board approval was obtained for conducting the animal study (IZ4-38). We assigned 10-week-old Sprague-Dawley male rats (*n* = 21) with an average weight of 305 g into study and control groups. Study group rats received β-TCP/PLLA/PGA and β-TCP nanobiomaterials on the right side of the mandible (Figure 2A). The experimental workflow of our study is described in Figure 2B. Aseptic conditions were followed during the surgical procedure. We used a 3-mixture anesthetic solution that consisted of medetomidine hydrochloride (0.15 mg/kg), midazolam (2 mg/kg), and butorphanol (2.5 mg/kg). The drugs were diluted appropriately with sterile water. Intraperitoneal deposition of the anesthetic solution in the animals was carried out. We painted and disinfected the external skin of the right submandibular region with povidone-iodine. A full-thickness skin incision (1 cm approx.) was made in the submandibular region, and sequential dissection of soft tissue layers such as subcutaneous tissue and muscle was performed. The periosteum was then incised to expose the buccal surface of mandibular bone at the angle region. We created a 4-millimeter-diameter bi-cortical critical size defect above the mandibular angle region as described earlier [17] on the right side using a drill bit and a trephine bur. The defect was then filled with β-TCP/PLLA/PGA and β-TCP nanobiomaterials as per the pre-operative plan. The Sham group animals did not receive any nanobiomaterials. Resorbable sutures were used for wound closure (Figure 2C). Antibiotic and analgesic injections were given after the completion of the surgery. The rats woke up from anesthesia about 2 h after surgery. They resumed normal activity and movement and had a good appetite. The weight and health condition of the rats were monitored regularly. All animals remained alive and healthy until the sacrifice period.

We sacrificed the animals at 3 timepoints: week 2, week 4, and week 12. Euthanasia was performed by inhalation of volatile anesthetic isoflurane in a closed glass chamber. Seven right mandibular specimens—3 from β-TCP/PLLA/PGA, 3 from β-TCP, and 1 Sham control—were collected at each time point. Post-sacrifice, the right hemi-mandible containing the nanobiomaterial was extracted and soaked in a 10% neutral buffered formalin solution in labeled containers.

### 2.3. Micro-Computed Tomography (Micro-CT) Procedure

The harvested rat mandibles were scanned using the Micro-CT scanner—CosmoScan FX (Rigaku Corporation, Tokyo, Japan). The scan time was 2 min. The voltage was 90 kV, and the current was 88 μA. The field of view was 10.24 × 10.24 × 10.24 mm, resolution 20 μm, and matrix 512 × 512 × 512. Calcium Hydroxyapatite (CaHA) phantoms containing 5 cylinders with varying densities of 0, 50, 200, 800, and 1200 g/cm^3^ (QRM, Moehrendorf, Germany) were scanned along with the specimens. The Micro-CT scanning was performed by In-Vivo Science Inc. (Kanagawa, Japan).

### 2.4. Estimation of Bone Volume to Total Volume Ratio (BV/TV), Inner Defect Diameter Reduction, and Bone Mineral Density (BMD)

Two individual researchers performed all analyses to avoid bias. The BV/TV ratio and inner defect diameter reduction were assessed with the aid of ImageJ (Fiji, version 2.14.0/1.54f) software. The plugin ‘TransformJ’ was used to correctly angulate the slides and view the defect region as a spheroid. For BV/TV calculation, we chose the Digital Imaging and Communication in Medicine (DICOM) slides containing the nanobiomaterials from anterior to posterior sections. The ‘Elliptical’ tool was used to draw a 4-millimeter circle to simulate the defect area. The selected slides formed a duplicate set of images containing only the region of interest (ROI). Thereafter, binary images were created. The ‘BoneJ’ plugin was used to compute the results showing the area/volume fraction from the ROI binary set (see Appendix A). The inner defect diameter for the middle slide in the Micro-CT dataset was calculated. We recreated the 4-millimeter defect on that particular slide and used the polygonal selection tool to trace the new bone within the confines of the defect. The marked area was measured using the ‘Measure’ option in the ROI Manager window.

BMD is defined as the volumetric density of CaHA in a biological tissue expressed as g/cm^3^. We estimated the BMD of new bone generated by the nanobiomaterials using CTAn version 1.19+ (Skyscan, Bruker, Kontich, Belgium). CTAn provides calibration of the BMD against either Hounsfield units (HUs) or attenuation coefficients (ACs). We chose to calibrate our specimens against the AC (mm^−1^), as it is a direct measure of X-ray absorption. Since the X-ray absorption of mineralized tissues is dominated by CaHA, the AC can be related to and compared to the mass density of that material [46]. Two of the five phantoms—800 and 1200 g/cm^3^ cylinders—were used to calibrate the AC. The formula used to calculate BMD was as follows: BMD (g/cm^3^) = AC − 0.435/−0.2675. After obtaining the formula, the DICOM Micro-CT data of the specimens were input into the CTAn software. The ROI chosen for measurement had a range of 50–70 slides, and the boundaries of the ROI were adjusted to include the defect size. A separate folder containing the ROI images was saved, and the BMD value was measured from the ROI. Three specimens from each group were used at each timepoint to assess the above-mentioned parameters.

### 2.5. Tissue Preparation: Immunohistochemistry (IHC) Staining

The scanned rat mandibular specimens stored in 10% neutral buffered formalin were prepared for IHC staining by employing the following steps: The specimens were demineralized using 10% ethylenediaminetetraacetic acid (EDTA) for 5 weeks. These specimens were then embedded in paraffin and made into blocks using a sealed automatic fixation and embedding device. The sectioning created 4-micrometer-thickness slices and contained the nanobiomaterials at the center, demineralized bone around, and soft tissue.

### 2.6. Hematoxylin–Eosin (H&E) and Immunohistochemical (IHC) Staining

All three specimens were stained in each group at every timepoint. Hematoxylin–Eosin (H&E) staining was performed for all the specimens, including the Sham control group. IHC staining was performed only for the study group specimens. The sections were deparaffinized with xylene and rehydrated with a series of ethanol solutions. They were then washed under water. Antigen activation treatment with citric acid buffer and EDTA was performed at 90 °C for 8 min. A phosphate buffered saline (PBS-0.01M, pH 7.4) wash was given, followed by a 3% hydrogen peroxide treatment for 5 min to quell endogenous peroxidase. Primary antibody staining was performed with anti-Runx2 rabbit polyclonal antibody (abcam: ab23981; 1:300 conc), anti-LEPR rabbit polyclonal antibody (Proteintech, 20966-1-AP; 1:150 conc), anti-human osteocalcin monoclonal antibody clone (BIO-RAD, 0400-0041; 1:3000 conc), and anti-Periostin rabbit polyclonal antibody (abcam: ab14041; 1:800 conc) at room temperature for 50 min. After another PBS wash, secondary antibody staining was performed with Histofine Simple Stain Rat MAX-PO (MULTI) (Nichirei Bioscience Inc., Tokyo, Japan 414191) at room temperature for 30 min. After another PBS wash, the slides were DAB-colored for 10 min and finally stained with Meyer’s hematoxylin for 30 s. The coloration was removed, a water wash was performed, and the specimen-containing slides underwent dehydration, permeabilization, and sealing. IHC staining was not performed for the specimens in the Sham group. All IHC protocols were conducted by SeptSapie. Co. Ltd. (Tokyo, Japan).

### 2.7. IHC Evaluation: Optical Density Assessment

Microscopic evaluation of the stained slides was conducted with a BX43 light microscope (Olympus Corporation, Tokyo, Japan) in concordance with the Olympus D21-CB digital photo system. ImageJ software was utilized for IHC assessments. We measured the IHC Optical Density (IHC OD) scores of Runx2, LepR, OCN, and Periostin using 3 specimens from both groups at each time point. Three images were taken from each specimen at 20× magnification. The plugin used for IHC OD assessment was the ‘IHC Profiler’. Using the DAB color deconvolution algorithm [47], the profiler automatically calculates a semi-quantitative score (see Appendix A). It can be converted into a quantitative score [48] with the following formula:

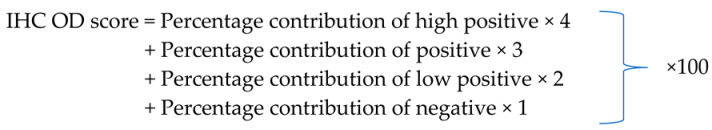



The average values were taken to exhibit the expression of the biomarkers in each group, making up the final IHC OD score.

### 2.8. Statistical Analysis

We used SPSS for Mac OS version 27.0 (IBM Corporation, Armonk, NY, USA) for statistical evaluation. Non-parametric tests such as Kruskal–Wallis and Mann–Whitney U tests were used to compare significance between the study groups—β-TCP/PLLA/PGA and β-TCP at weeks 2, 4, and 12. A Wilcoxon–Signed rank test was used to compare the statistical differences of the same study group across different timepoints. A *p*-value < 0.05 indicated statistical significance. We used the post hoc Bonferroni test to decrease error probability.

## 3. Results

### 3.1. New Bone Assessment

The new bone regeneration in relation to both scaffolds was assessed using the Micro-CT data obtained from rat mandibular specimens at weeks 2, 4, and 12 (Figure 3A). The new bone was macroscopically visible on the outer surface, in-between the fibers of β-TCP/PLLA/PGA, and within the pores of β-TCP. The volume of new bone increased steadily, and solid bone growth from the rim of the defect to the center was appreciable by week 12 in both groups. The Sham group did not show much difference at weeks 2 and 4. However, small bony stumps could be seen extending from the inner rim of the defect by week 12. Both groups showed good radiographic visibility. Since β-TCP has a composition similar to that of natural bone, it appeared consistent with the host bone in certain sections of the Micro-CT images (Figure 3A).

#### 3.1.1. BV/TV Results

The BV/TV ratio between both groups was comparable at each time point. BV/TV results showed slightly less bone regeneration in the β-TCP/PLLA/PGA group than in the β-TCP group; this was statistically significant by weeks 4 and 12. It was also observed that the new bone formed varied significantly between the same group at each time point. By week 12, the β-TCP/PLLA/PGA group showed a sharp decline in new bone formation, owing to lesser β-TCP content and a general saturation of the bone regeneration process (Figure 3B).

#### 3.1.2. Inner Defect Diameter Reduction

We noticed a gradual reduction in the created 4-millimeter defect size at weeks 4 and 12 in the case of both β-TCP/PLLA/PGA and β-TCP (Figure 3C). It was interesting to note that the bone outgrowth from the defect margin in β-TCP/PLLA/PGA was greater than that of β-TCP in weeks 2 and 4, with a significant difference in week 4. β-TCP showed a significant rise in defect closure between week 2 and week 4. The β-TCP group showed higher closure of the inner defect bone than β-TCP/PLLA/PGA in week 12. Though the volume of bone formed in the β-TCP group was greater by week 12, this difference was statistically insignificant (Figure 3D).

#### 3.1.3. BMD Results

At every time point, the BMD of trabecular bone formed in accordance with the β-TCP/PLLA/PGA group was slightly less than the β-TCP group, but the results were comparable; the difference between the groups was significant only at week 12. The BMD comparison between the same group with regard to both β-TCP/PLLA/PGA and β-TCP at weeks 2, 4, and 12 was significant. This shows continuous and consistent bone regeneration and maturation in both groups at every stage (Figure 3E).

### 3.2. H&E Staining

Stained H&E slides displayed the amount of newly regenerated and matured bone at each time point (Figure 4). The bony edges of the defect showed intense bioactivity in both groups, mainly at week 2, which then reduced gradually at week 4, with minimal to no reaction in week 12. In the β-TCP/PLLA/PGA group, the bioactive reaction was seen to surround the fibers, and in the β-TCP group, numerous cells were seen within the pores. Scattered mature bone regeneration areas were seen in the β-TCP/PLLA/PGA group by week 12, whereas dense bony islands seemed to replace the defect region in the β-TCP group. No intense inflammatory reaction was observed in either group, and no multinucleated cells were observed at locations surrounding the scaffolds. The Sham group did not show many changes; mild inflammation was seen at week 2. The defect region in the Sham group was later occupied by muscle tissue at the end of week 12 (Figure 4).

### 3.3. IHC Analyses—Expressions of Different Biomarkers

#### 3.3.1. Runx2 Expression

Cells positive for Runx2 were expressed abundantly in both groups at weeks 2 and 4. Though the expression gradually tapered at the end of week 12, β-TCP/PLLA/PGA showed better expression than the β-TCP group (Figure 5A). Runx 2 accumulation was noted adjacent to the periosteum and alongside new bone that was forming.

The IHC OD score for Runx2-positive cells was higher in the β-TCP/PLLA/PGA group than the β-TCP group at week 4. The scores in both groups were comparable in week 2, with a mild increase in the β-TCP group. However, at week 12, the score of β-TCP/PLLA/PGA was significantly higher than β-TCP, indicating the bioactive potential of β-TCP/PLLA/PGA even at later stages. IHC OD scoring showed significance at different time points for the same group (Figure 5B).

#### 3.3.2. LepR Expression

LepR expression was vivid in both groups, markedly higher at week 4 in the β-TCP/PLLA/PGA group and at weeks 2 and 12 in the β-TCP group. The LepR-positive cells were seen lining the parent bone-scaffold interface at the defect margins and near the periosteum. There was a strong expression of these cells interspersed between and in close contact with the β-TCP/PLLA/PGA fibers, and the positive cells were expressed mainly around the scaffold in the β-TCP group. The antibody-positive cells were also seen scattered around newly formed bone. It was very evident that LepR expression followed the same pattern as observed with Runx2 expression in both groups across time points (Figure 6A).

The β-TCP/PLLA/PGA group showed the highest score with statistical significance at week 4 among all time points. The score of the β-TCP/PLLA/PGA group at week 12 decreased more significantly, but it was still higher than β-TCP. The β-TCP group had a significantly reduced IHC OD score between weeks 2 and 12 (Figure 6B).

#### 3.3.3. OCN Expression

In the β-TCP/PLLA/PGA group, week 2 displayed an OCN deposition closer to the host bone region. Due to bone maturation at week 4, the expression was marked. By week 12, OCN was exclusively seen around the laid-down lamellar bone, but the intensity seemed to be slightly less in comparison to the β-TCP group. At week 2, intense deposition of OCN antibodies was seen closer to the defect margins in the β-TCP group. Intrapore expression of OCN was evident in β-TCP by week 4. At week 12, the highest expression was observed in the defect space around the mature bony islands (Figure 7A).

The IHC OD scoring of OCN showed comparable results, with both groups being nearly equal at weeks 2, 4, and 12 with mild changes. The β-TCP group had a higher statistical increase in scoring than the β-TCP/PLLA/PGA group at week 2. Week 4 scoring was significantly higher in β-TCP/PLLA/PGA than week 2 (Figure 7B).

#### 3.3.4. Periostin Expression

Marked periostin activity was seen in both the β-TCP/PLLA/PGA and β-TCP groups at weeks 2 and 4. The β-TCP/PLLA/PGA group had periostin deposited directly around them following the fiber architecture, while in the β-TCP group, the protein was seen in the pore region as well as around the scaffold. The β-TCP/PLLA/PGA group showed more intense periostin staining at weeks 2 and 4 than the β-TCP group, which was reduced by the end of week 12 (Figure 8A).

The IHC OD scoring for periostin showed interesting results. Periostin was found to be significantly increased in β-TCP/PLLA/PGA at both weeks 2 and 4, but the scores were reduced by week 4. An insignificant decrease in periostin score was seen by week 12 in the β-TCP/PLLA/PGA group. The β-TCP group had a more significant reduction in scores by week 12 than at week 4 (Figure 8B).

## 4. Discussion

The ‘Triangular concept’ of bone regeneration well iterates the basic requirements that are considered mandatory for bony healing, as in osteoconductive scaffolds, growth factors, and osteogenic cells. This was modified to include the mechanical environment, a fourth and prime factor encompassing vascularity and host conditions, making up the ‘Diamond concept’, proposed by Giannoudis et al. [49]. The mechanical environment of the graft should be considered just as important as the biologic graft properties for the implanted graft to be successful [50]. Scaffolds have been classified according to their manufacturing and surface characteristics into the following types: (a) fibrous, (b) hydrogel, (c) microsphere, (d) composite, (e) acellular, and (f) porous, among other modifications [51]. Prominent improvements to conventional scaffold fabrication are needed in lieu of discrepancies concerning hydrophilic nature, bioactive tendency, and degradation. Some of the vital factors to be considered are morphology, aperture, porosity, and spatial orientation. The right amount of pore size is advantageous toward promoting cell attachment, migration, proliferation, differentiation, and finally bone regeneration [52]. Technical issues can affect biomaterial functionality [32] and can potentially alter the microenvironment relative to the scaffold, thereby causing cell infiltration. We used a novel nanobiomaterial, β-TCP/PLLA/PGA, in the form of electrospun fibers to assess the efficacy of bone regeneration and bioactive potential in lieu of its excellent handling properties as a replacement for conventional β-TCP for utilization in maxillofacial bone defects.

### 4.1. Bone Regenerative Capability of β-TCP/PLLA/PGA in Comparison to β-TCP

The presence of the majority of Ca/P in both scaffolds made it easier to demarcate them in the Micro-CT sections. BV/TV is a reliable method to indicate the new bone formed with respect to scaffolds. It is generally noted that bone regeneration after significant trauma/injury proceeds in an orderly fashion, with inflammation and progenitor cell recruitment at the initial phases, followed by bone deposition and maturation at later stages [53,54]. Both β-TCP/PLLA/PGA and β-TCP showed mature regenerated bone and Runx2 and LepR expressions, though decreased, at week 12. This shows that bone deposition with the application of these scaffolds continues even at a later stage, as observed in previous studies [55]. Osada et al. stated that bioactivity and new bone formation were present even by week 12 after biomaterial implantation [56]. This demonstrates the osteogenic potential of the β-TCP/PLLA/PGA nanobiomaterial at later periods when new bone regeneration may reach saturation [41,56]. BV/TV results from our study showed that the new bone formed in the β-TCP/PLLA/PGA and β-TCP groups were comparable, except at week 12. It is important to note that the bone regenerated by β-TCP/PLLA/PGA was comparable to β-TCP scaffold at weeks 2 and 4, even when it consisted of only 70% β-TCP by weight. With β-TCP/PLLA/PGA, new bone extended from the defect margins toward the center. This feature was particularly proven with our analysis of inner defect diameter, which significantly started to reduce by week 2 in the β-TCP/PLLA/PGA group. The new bone in the β-TCP group extended directly from the ends of the host bone defect and inside the pores. The minimum intrapore size for successful bone regeneration is approximately 50–100 μm [57], as this size is necessary for osteoblasts to proliferate [55]. Pore interconnections must be above 50 μm [57] to allow bony ingrowth; our material satisfied the above criteria. It was evident that both nanobiomaterials showed competitive BMD scoring. Cell attachment to the scaffold is directly proportional to the surface roughness of the nanobiomaterial. In the case of electrospun fibers, though the inner surface is extremely rough, the outer surface may not possess this property [58,59]. This explains the increased bioactivity and spurt in bone growth with the β-TCP/PLLA/PGA group that occurred following initial degradation in week 4 compared to week 2. Surface coating has been shown to improve attachment and hydrophilicity to enable better outcomes [59]. β-TCP has a highly rough surface and allows easy cell proliferation [17]. The Sham group was devoid of significant changes by the end of week 12, demonstrating that our critical-sized bone defect could not heal without intervention.

#### Importance of Calcium and Phosphate Ions in Aiding Bone Regeneration

The presence of ionic calcium is a pre-requisite to induce favorable effects concerning bone regeneration. The osteoconductive effects of β-TCP bone substitute can be attributed to the formation of an apatite layer that has been observed to form upon immersion in ionic environments [11], thus facilitating the release of calcium ions. Efficient degradation of β-TCP from both nanobiomaterials releases large amounts of free Ca^2+^ and PO_4_^3−^ ions, which creates the necessary space for tissue ingrowth [60]. The released Ca^2+^ ions interact with the ECM and affect the adsorption of ECM proteins, thereby regulating cell adhesion and tissue growth [61]. Ca^2+^ ions cause nitric oxide release and stimulate osteogenic cells for osteoid deposition [62]. Ca^2+^ ions also activate the ERK1/2 pathway and the PI3K/Akt pathway to increase the life span of osteoblasts [63]. Ca^2+^ ions control osteoclast proliferation and bone resorption [64]. Ca^2+^ ions promote proliferation and motility of vascular endothelial cells and activate factors for endothelial cell proliferation [65]. In mice, 2–4 mmol/L Ca^2+^ ions aid proliferation and survival of primary osteoblastic cells; in 6–8 mmol/L concentration, they help in osteoblast differentiation and matrix mineralization [66]. In humans, 14 mmol/L Ca^2+^ ions are needed to maintain a round osteoblast morphology and encourage osteoblastic cell differentiation [66]. Hence, Ca^2+^ ions play an important part in bone regeneration and maturation. In addition, PO_4_^3−^ ions play a major role in the regulation of the IGF-1 and ERK1/2 pathways, which influence osteoblast differentiation and growth [67]. PO_4_^3−^ ions have also shown negative feedback with the RANK ligand and inhibit bone resorption through osteoclast inhibition [68].

Pure β-TCP has a longer resorption time [17]; the material has been observed in vivo 2.5 years after implantation [69]. The amount of β-TCP is directly proportional to the degradation rate; an increased amount of β-TCP in turn increases the time taken for degradation [31]. In contrast, PLLA/PGA degrades faster, and this factor can be controlled by altering the ratio of PLLA and PGA. With regard to β-TCP/PLLA/PGA degradation, we hypothesize that PLLA/PGA degraded quickly and released β-TCP more efficiently. This explains the comparable new bone formation and bioactivity seen in both groups during the initial stages. As the availability of β-TCP in β-TCP/PLLA/PGA decreased in later stages due to less content, there was a decline in bone formation by week 12. As nanobiomaterial degradation eventually ensues, the defect space previously occupied by the scaffold is replaced by new bone [40]. This was also found in our results by week 12. β-TCP-based nanobiomaterials have been credited for their excellent biocompatibility, which was true in our study as well; we did not notice any inflammation/fibrous tissue formation in either group. PLLA/PGA degradation can create an acidic environment [70] owing to the PLLA component, but the addition of β-TCP neutralizes the acid byproducts [71]. For osteoblasts to differentiate, a pH value of normal to mildly alkaline (7.4 to 8.6) is mandatory [72]. This factor is not an issue when β-TCP is combined with PLLA/PGA, as PLLA/PGA has reduced and transient inflammatory effects; it degrades faster and completely (end products are CO_2_ and H_2_O) without leaving any residual toxic by-products.

Our speculation of the method of bone regeneration is depicted in Figure 9. Fast degradation of pure β-TCP and quick resorption of PLLA/PGA cause the release of large amounts of Ca^2+^ and PO_4_^3−^ ions in the defect site. This attracts progenitor cells from the host bone marrow as well as from the circulation to convert and proliferate into osteogenic cells. As a response to injury, skeletal stem cells (SSCs) from the periosteum covering the buccal aspect of the scaffold convert into osteogenic lineage cells. Simultaneously, angiogenesis that proceeds in early stages due to the influence of Ca^2+^ ions is an important requisite for bringing forth faster healing. The gathered progenitor cells differentiate and deposit osteoid tissue conveniently in the scaffold matrix. We deduce that the initial prompt local response is a key factor in the induction of bone regeneration. 

### 4.2. Significance of Biomarker Expression during Bone Regeneration

An adequate amount of biomarker expression equates to active bone regeneration. It has been reported that β-TCP can cause an increase in the levels of osteoblast transcription factors such as Runx2 and other differentiation biomarkers such as collagen type-1, ALP, OPN, and OCN even when supplemental factors essential for bone regeneration are not applied [73,74]. It is well known that Runx2 absence can inhibit osteoblastogenesis [75], and it is necessary for osteoblast lineage commitment from MSCs [76]. Runx2 levels decrease when osteoblasts reach the maturation stage [77]. The MSCs and progenitor cells are considered LepR-positive cells [78]. Yang et al. detected Runx2-GFP+ in osteoblasts and osteocytes, thus indicating the derivation of Runx2-positive cells in osteoid tissues as well as in bone marrow [78]. Thus, it can be assumed that Runx2 expression is a direct reflection of osteoblastic activity and LepR in stem cells. Interestingly, our results showed that Runx2 and LepR expressions followed a similar expression trend, and the IHC OD scoring mirrored the same finding. Both Runx2 and LepR expressions reached a peak in the β-TCP/PLLA/PGA group at week 4 and decreased by week 12. However, their expression was higher than that of the β-TCP group and was still pronounced by week 12. These findings are in agreement with previous literature [35]. The attachment of bioactive cells directly to the surface of β-TCP/PLLA/PGA could be due to the surface interspersion of β-TCP particles. Higher expression of OCN is seen in the mineralized matrix rather than in the initial bone regeneration stages [79]. The OCN expression follows that of Runx2 [23]. We found a parallel OCN increase in the β-TCP/PLLA/PGA group at week 4. The OCN staining at week 12 was comparable in both groups, indicating the maturation of newly formed bone. These results, without doubt, prove that the bioactive abilities of β-TCP/PLLA/PGA are comparable to β-TCP, with only 70% weight of β-TCP. From our observation, the expression of Runx2 and LepR is highest during the inflammatory phase following injury as a consequence of active progenitor cell recruitment. OCN expression during the initial stages is almost negligent; positive staining for OCN has been noted in the late stages following maturation of the deposited bone tissue. By week 4, concurrent with β-TCP degradation, Runx2 and LepR expressions reached a plateau, especially in the β-TCP/PLLA/PGA group, which highly indicates steady bone regeneration. As a result of subsequent saturation in the regeneration process, active progenitor cell recruitment is decreased, causing a decline in biomarker levels by week 12.

Periosteum is an integral part of every osteoid tissue. It is highly vascular, provides nutrition, responds to mechanical stress, and aids in the healing of the bone underneath by supplying SSCs, making it a vital component of bone regeneration [80,81]. Periostin is a 90-kDA matrix protein secreted by periosteal osteoblasts; it regulates bone homeostasis and can induce ectopic osteogenesis in vivo [82]. Periosteum gives rise to periosteal cells (PCs) that can differentiate into osteogenic, adipogenic, and chondrogenic cell lineages [82]. It has been shown that SSC recruitment happens locally, and bone marrow stem cells indirectly aid osteogenesis by supplying growth factors and do not participate in active bone regeneration at later stages [83]. Periostin is initially found in the Cambium layer of activated periosteum and then migrates to the periphery of the callus at later stages of healing [82].

We conducted periostin evaluation to assess the source of progenitor cells and osteogenic cells in our study. Abundant expression of periostin was seen at both weeks 2 and 4. Periostin expression was reduced by week 12, due to less recruitment of SSCs, but still remained slightly higher in the β-TCP/PLLA/PGA group than the β-TCP group. Deposition was noted within the pores of β-TCP but followed the fiber pattern of the β-TCP/PLLA/PGA group. Staples et al. studied the effect of electrospun mats on periodontal regeneration and noted that periostin deposition follows the fiber architecture, as in our study [43]. This feature can be attributed to the fact that electrospun materials possess an orderly fiber shape bearing resemblance to ECM, thereby enhancing tissue maturation. Periostin deposition around the fibers is an indication of osteogenic activity and bone deposition around the fibers. The precise molecular mechanisms concerning the above-mentioned are unclear, and further research is required [43].

### 4.3. Limitations

We could not assess angiogenesis biomarkers during bone regeneration in our study. Since this is a preliminary analysis, we used a limited number of rats per group in our study; the analysis of more specimens could greatly benefit the outcome of statistical testing. We are working to overcome these limitations in our future research work with the novel nanobiomaterial.

### 4.4. Future Perspectives

Third-generation (HA/PLLA) and fourth-generation (HA/PLLA/PGA) bioresorbable materials can be combined with β-TCP through electrospinning by controlling the compositions to enhance the performance of each constituent component. Combination of the cotton-like β-TCP/PLLA/PGA in addition to stem cells is an interesting future prospect that could have interesting clinical results when explored.

## 5. Conclusions

A thorough comparison of the novel material β-TCP/PLLA/PGA and the conventional material β-TCP revealed comparable results between the two materials in terms of bone regeneration and bioactivity, even when β-TCP/PLLA/PGA had a lesser content of β-TCP. β-TCP/PLLA/PGA showed higher expressions of Runx2 and LepR at later time periods as well, indicating its potentiality to attract and aid proliferation of osteogenic cells from both bone marrow and periosteum-derived sources better than β-TCP block. β-TCP particles on the surface of β-TCP/PLLA/PGA resulted in bioactive cell proliferation around the fibers. High periostin expression during the early and late stages could be attributed to its novel fiber shape.

As per our hypothesis, β-TCP/PLLA/PGA showed good bone regenerative capability brought about by the induction of bioactive/osteoconductivity, as explained above. We conclude by stating that the electrospun nanobiomaterial β-TCP/PLLA/PGA has osteoinductive and osteoconductive effects with ease of handling, is useful in variable quantities as an efficient filling material, and is applicable to a variety of maxillofacial bone defects.

## Figures and Tables

**Figure 1 nanomaterials-14-00091-f001:**
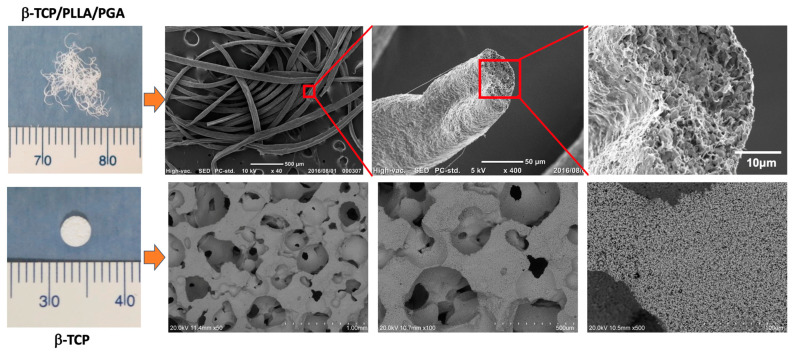
β-TCP/PLLA/PGA (up) and β-TCP (down) scaffolds. β-TCP/PLLA/PGA electrospun cotton-like fibers (**upper left**) and SEM images of the fibers (**from left to right**): ×50, ×400, and ×2000 magnifications. β-TCP bone substitute block (**lower left**) and SEM images of β-TCP block (**from left to right**): ×50, ×100, and ×500 magnifications.

**Figure 2 nanomaterials-14-00091-f002:**
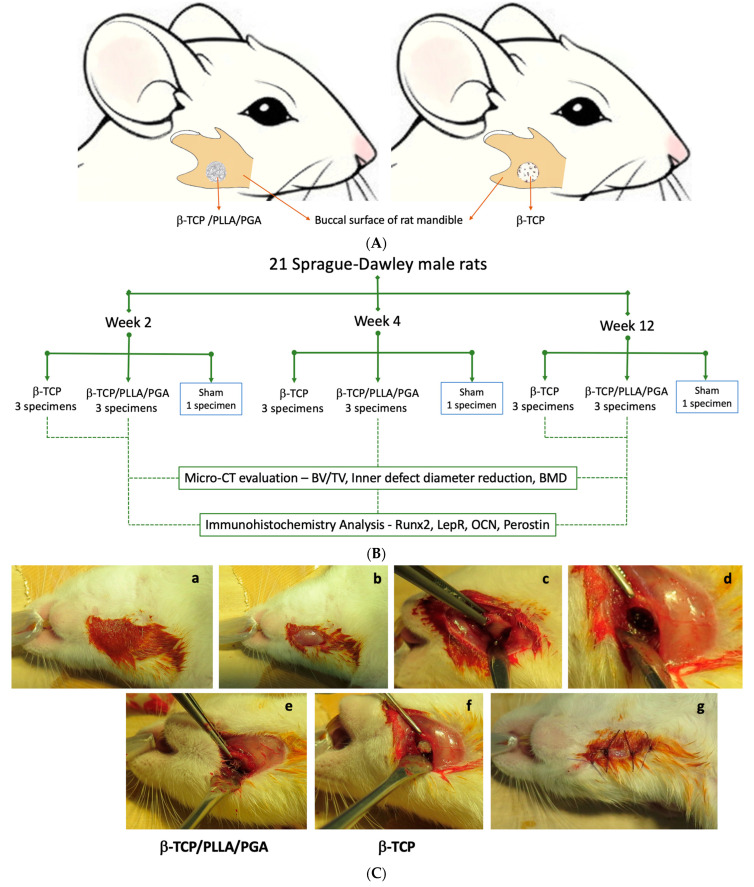
(**A**): Animation of the rat mandible showing scaffold placement inside the mandibular critical defect (4 mm) on the right side. (**B**): Flowchart detailing experimental workflow. (**C**): Steps during surgery: (**a**) Disinfection of the rat submandibular region; (**b**) Skin incision exposing the muscular layer; (**c**) Periosteal incision and mandibular angle region exposed; (**d**) Creation of a 4-millimeter defect above the mandibular angle; (**e**,**f**) β-TCP/PLLA/PGA and β-TCP packed into the defect, respectively; (**g**) Wound closure.

**Figure 3 nanomaterials-14-00091-f003:**
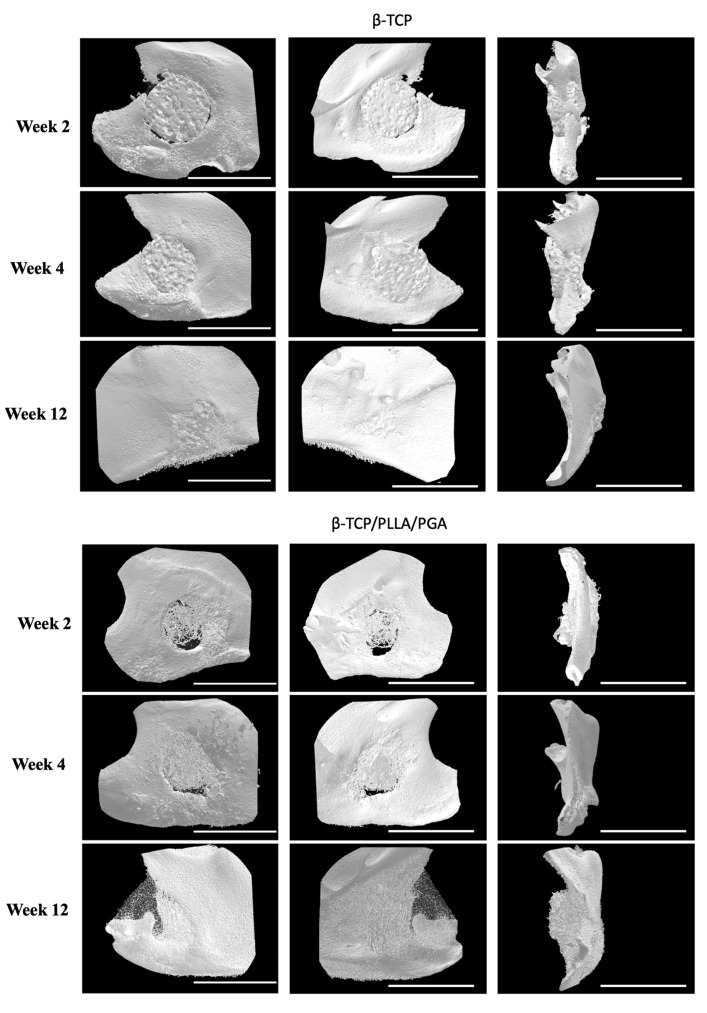
(**A**): 3D reconstructed Micro-CT images displaying buccal, lingual, and sagittal views (**from left to right**). Comparison of each group at weeks 2, 4, and 12. Note the subsequent defect rim closure progressing over the weeks. The Sham group had negligible bone regeneration, with small stumps visible by week 12. A defect size of 4 mm was standard across all groups. Volume of biomaterial applied: β-TCP/PLLA/PGA group—3 mg, and β-TCP group—16 mg. Scale bar = 1 mm. (**B**): Graph demonstrating the BV/TV ratio between β-TCP/PLLA/PGA and β-TCP. The BV/TV ratio was consistently comparable between both groups at set time points. A defect size of 4 mm was standard across all groups. Volume of biomaterial applied: β-TCP/PLLA/PGA group—3 mg, and β-TCP group—16 mg (graph symbols denote statistical significance). * *p* ≤ 0.05; ** *p* ≤ 0.01. (**C**): Illustration of new bone formation and defect diameter reduction in the β-TCP/PLLA/PGA group. The defect diameter was recreated (yellow circle), new bone formed within the confines of the defect was traced (blue marking), and the corresponding volume was drawn. (**D**): New bone growth from the inner edges of the defect. β-TCP/PLLA/PGA had faster defect rim closure than β-TCP at weeks 2 and 4 (graph symbols denote statistical significance). * *p* ≤ 0.05. (**E**): Differences in the BMD of trabecular bone formed in both groups were analyzed using CTAn software version 1.19+ (graph symbols denote statistical significance). * *p* < 0.05; ** *p* ≤ 0.01.

**Figure 4 nanomaterials-14-00091-f004:**
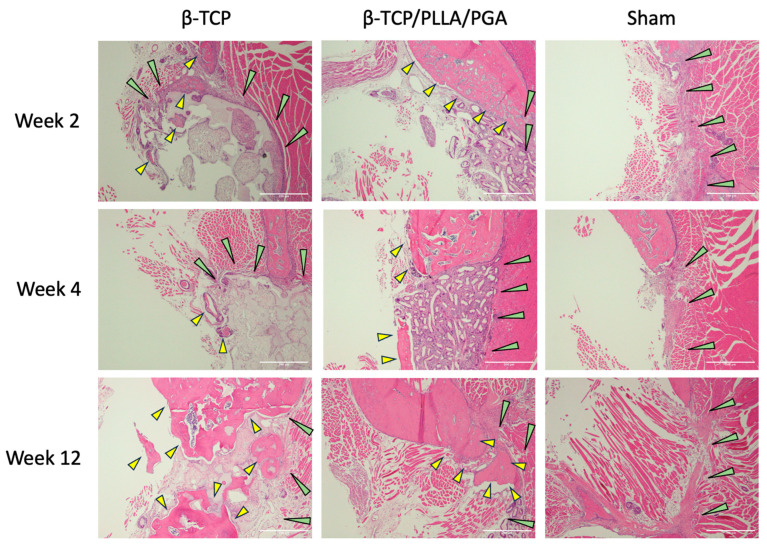
H&E view of β-TCP/PLLA/PGA and β-TCP groups at weeks 2, 4, and 12. Increased activity was observed with regard to the defect margins. Green arrows depict the defect margin; yellow arrows denote newly formed bone. Note the absence of inflammation in both groups, especially at initial time points (×4 magnification, scale bar = 200 μm).

**Figure 5 nanomaterials-14-00091-f005:**
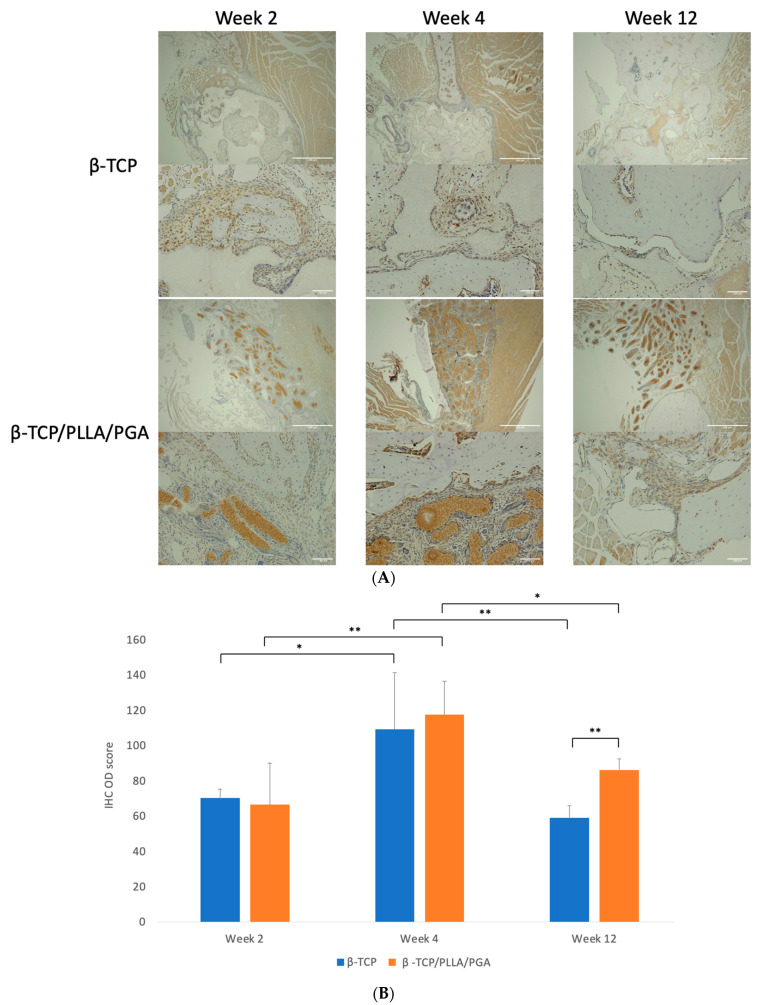
(**A**): Runx2 expression in β-TCP/PLLA/PGA and β-TCP groups (above: ×4 magnification, scale bar = 200 μm; below: ×20 magnification, scale bar = 100 μm). (**B**): Variance in the Runx2 expression. The β-TCP/PLLA/PGA group had significantly high expression at the end of week 12 (graph symbols denote statistical significance). * *p* < 0.05; ** *p* ≤ 0.01.

**Figure 6 nanomaterials-14-00091-f006:**
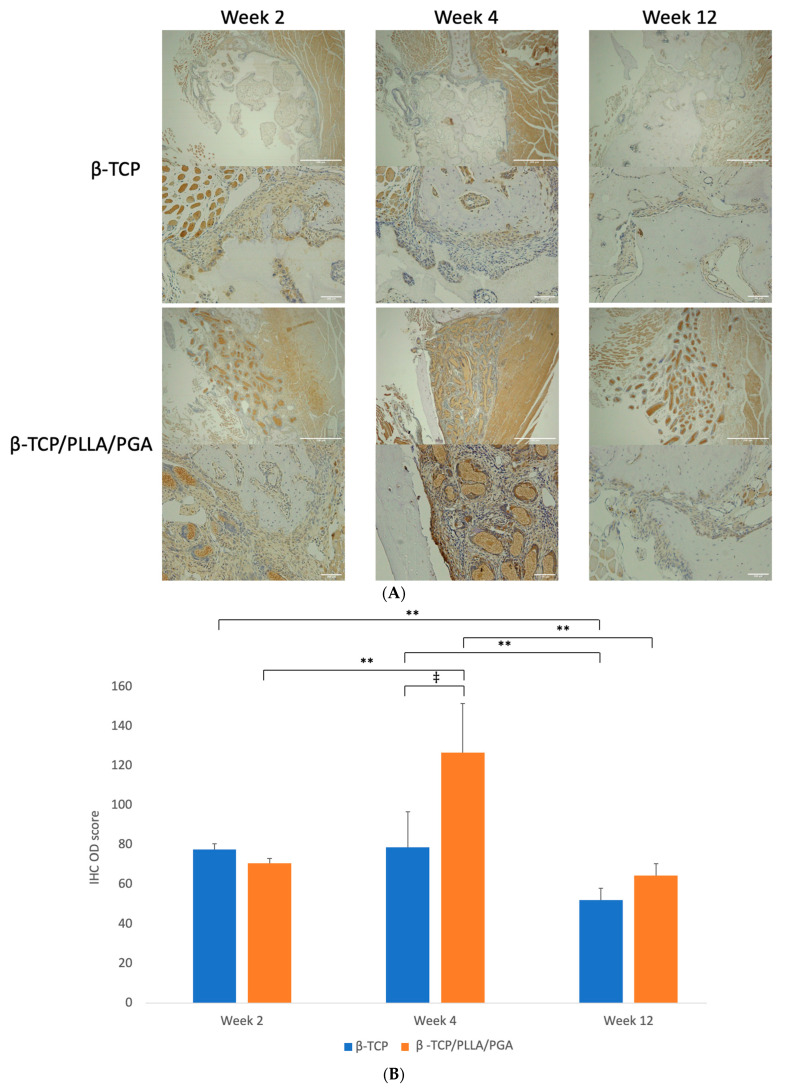
(**A**): Image showing LepR-positive cells in reaction to the implanted scaffold. Strong expression of LepR-positive cells in close contact with the β-TCP/PLLA/PGA group by week 4 (above: ×4 magnification, scale bar = 200 μm; below: ×20 magnification, scale bar = 100 μm). (**B**): Graph depicting the LepR IHC OD score. Results were parallel to Runx2 scores (* graph symbols denote statistical significance). ‡ *p* < 0.005; ** *p* ≤ 0.01.

**Figure 7 nanomaterials-14-00091-f007:**
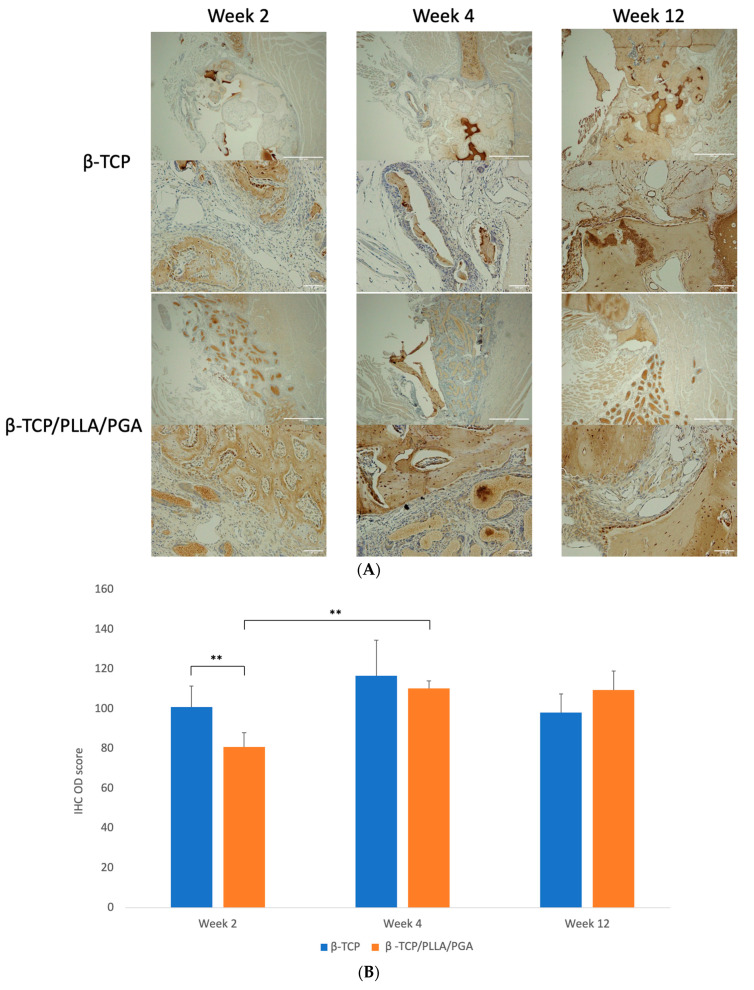
(**A**): OCN expression as seen at weeks 2, 4, and 12. β-TCP had consistent levels at weeks 4 and 12, indicating an increase in mature regenerated bone (above: ×4 magnification, scale bar = 200 μm; below: ×20 magnification, scale bar = 100 μm). (**B**): OCN–IHC OD scoring graph. Comparable results were observed at all time points (* graph symbols denote statistical significance). ** *p* < 0.01.

**Figure 8 nanomaterials-14-00091-f008:**
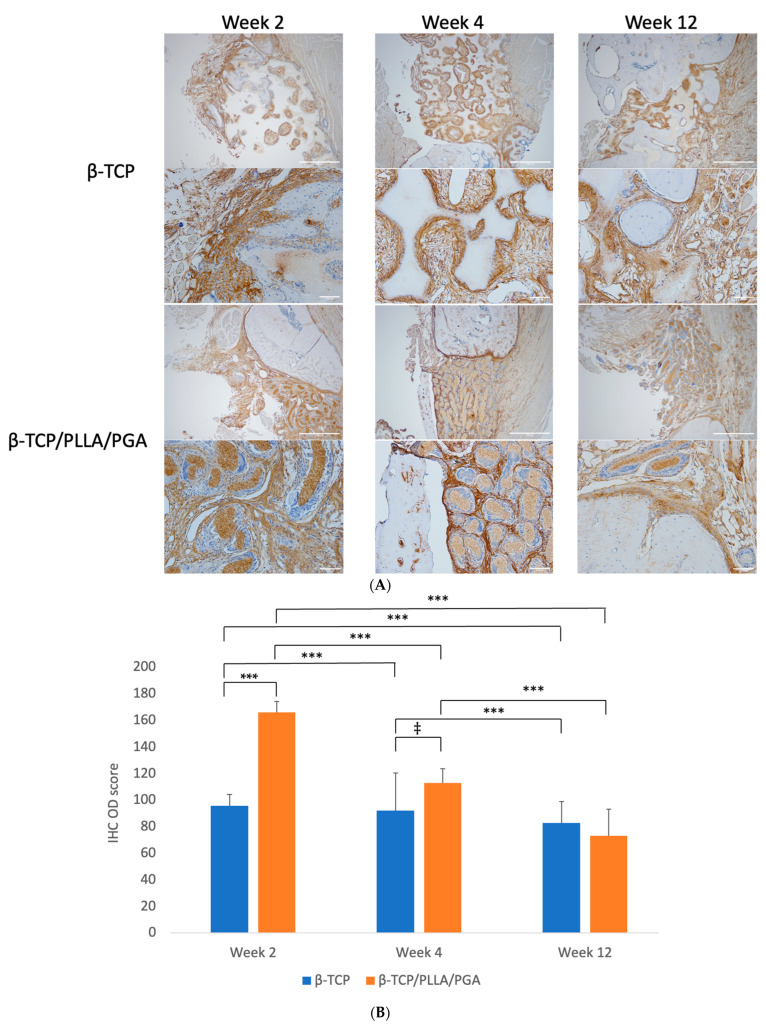
(**A**): Periostin deposition around the scaffolds. The protein deposition followed the shape of β-TCP/PLLA/PGA fibers, evident at week 4 (above: ×4 magnification, scale bar = 200 μm; below: ×20 magnification, scale bar = 100 μm). (**B**): Periostin IHC OD values. Highest scoring seen in β-TCP/PLLA/PGA at week 2 and thereafter at week 4 (* graph symbols denote statistical significance). ‡ *p* ≤ 0.005; *** *p* ≤ 0.001.

**Figure 9 nanomaterials-14-00091-f009:**
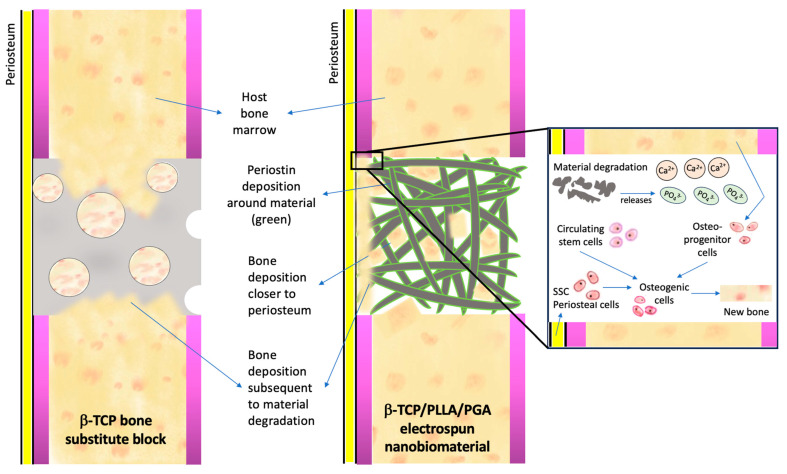
Mechanism of bone regeneration induced by calcium–phosphate scaffolds. Please refer to the text above for an explanation.

## Data Availability

Data are contained within the article and Appendix A.

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
