# Peer review of "In Vivo Evaluation of Bone Regenerative Capacity of the Novel Nanobiomaterial: β-Tricalcium Phosphate Polylactic Acid-co-Glycolide (β-TCP/PLLA/PGA) for Use in Maxillofacial Bone Defects"

_nanomaterials, 2023, doi:10.3390/nano14010091_

Round 1

Reviewer 1 Report (Previous Reviewer 1)

Comments and Suggestions for Authors

Specific comments

1) Please add some in vitro experiment results; before concluding on the successful bone regeneration of the new platform incorporating PLLA and PGA in β-TCP, it is necessary to evaluate the platform qualitatively and quantitatively.

2) If the defect was treated with the same 4 mm diameter in all groups, it is needed to revise the data in figure 8 because the addition of PLLA/PGA does not match the trend of bone regeneration.

3) Compared to the degree of regeneration in weeks 2 and 4, the tendency of the degree of regeneration in week 12 is completely reversed. In figures 8 and 9, the data, especially from week 12, does not seem to be sufficient data to support the author's claim.

4) In the LepR Expression section, the IHC OD score of the β-TCP/PLLA/PGA group at week 12 decreased ‘more significantly’ than the β-TCP group. Please revise the manuscript.

5) It is recommended that a general or detailed explanation of electrospinning technology be added to the introduction rather than a discussion.

6) Please present if there is bone regeneration data analyzed between 6 and 12 weeks. The author's in vivo experiment does not seem to agree with the results of Osada et al.

7) Please combine figure 16 into one figure. Also, please change the first figure to an image that highlights the regenerative ability of the β-TCP/PLLA/PGA group.

8) Please add a description of the black block in figure 16 or remove the black block.

9) Please add the sham data in all of the graph and compare with β-TCP and β-TCP/PLLA/PGA group.

Comments on the Quality of English Language

.

Author Response

Comments 1: Please add some in vitro experiment results; before concluding on the successful bone regeneration of the new platform incorporating PLLA and PGA in β-TCP, it is necessary to evaluate the platform qualitatively and quantitatively.

Response 1: Thank you for the above comment. The material tested is commercially available in the market under the names – ReboSorb®, ReBOSSIS-J®, and ReBOSSIS-MT® for clinical use in human patients. Extensive in-vitro testing has been performed to rule out the possibility of cytotoxicity and also the potentiality to attract and allow osteogenic cell proliferation. This material has also been FDA approved for clinical use since the year 2014. The in-vitro effects are of the commercially available material are well established and the composition is standardized. Many other specialties have conducted in-vivo research analysis on the behavior of the material in various defect models across the world, the details of which have been added to the main manuscript under the Introduction section from Line 147. Electrospun cotton-like β-TCP/PLLA/PGA fibers were initially developed as a bone substitute for placement in dental defects and around implant devices. They have also been employed in periodontal regeneration with good results. There has been no in-vivo research study that reports the effectiveness of the material in-vivo with regard to new bone regeneration in maxillofacial bone defects, to the best of our knowledge. Hence, our research paper focused on conducting and reporting an in-vivo study in the maxillofacial region, comparing β-TCP/PLLA/PGA nanobiomaterial with the conventional standard, β-TCP bone substitute block. We have added a few in-vitro characteristics of β-TCP/PLLA/PGA nanobiomaterial in the Introduction section from Line 131 for further enunciation of material properties.

Comments 2: If the defect was treated with the same 4 mm diameter in all groups, it is needed to revise the data in figure 8 because the addition of PLLA/PGA does not match the trend of bone regeneration.

Response 2: We thank you for the given comment. From our understanding of the reviewer’s question, we would like to explain as follows: the addition of PLLA/PGA material to β-TCP bone substitute was to solely improve the mechanical characteristics of the electrospun fibers. β-TCP has been combined with various other materials by previous researchers. We chose to assess a material that combines β-TCP with PLLA/PGA owing to its processability, biocompatibility, and the ability to alter the degradation time and mechanical properties by altering the ratio of components. Though the PLLA in PLLA/PGA can result in production of acidic byproducts during degradation, this environment is neutralized by β-TCP. PLLA/PGA is devoid of any bioactive/ osteoconductive property; it improves handling characteristics and provides adequate mechanical strength. Our previous research comparing PLLA/PGA with materials containing HA showed less new bone formation associated with PLLA/PGA. The new bone regenerated and the osteoinductive/ osteoconductive effects exhibited for the same are brought about only by β-TCP bone substitute in our current research. The trend of bone regeneration seen in our results is due to faster degradation of PLLA/PGA, thus releasing more amounts of β-TCP in the milieu. Plain β-TCP takes a longer time to degrade (2.5 years in-vivo). Hence, we would like to state that the data presented in Figure 8 is by far correct to the best of our knowledge. The above explanation has been added to the Discussion section from Line 596.

Comments 3: Compared to the degree of regeneration in weeks 2 and 4, the tendency of the degree of regeneration in week 12 is completely reversed. In figures 8 and 9, the data, especially from week 12, does not seem to be sufficient data to support the author's claim.

Response 3: Thank you for the comment. We would like to re-iterate that the quantity of β-TCP used in β-TCP/PLLA/PGA nanobiomaterial (3mg) is lesser than that used in the β-TCP bone substitute block (16mg). It has been shown that in-vivo degradation of pure β-TCP takes a longer time. PLLA/PGA on the other hand, degrades much faster in the in-vivo setting. It has been shown that biodegradable materials combined with PLLA/PGA as a result of its faster degradation, decouple and get released in the vicinity quickly. We hypothesize that the above-explained mechanism is same for the β-TCP/PLLA/PGA nanobiomaterial. Due to increased amount of β-TCP present in the milieu, bone regeneration at initial stages is comparable to that of β-TCP bone substitute block; at later stages due to general saturation of the regeneration process and potentially lesser amounts of β-TCP release, the bone regeneration is seen to decline by week 12. The explanation concerning degradation of both the materials and its effect on new bone regeneration at later stages has been added under the Discussion section from Line 593.

Comments 4: In the LepR Expression section, the IHC OD score of the β-TCP/PLLA/PGA group at week 12 decreased ‘more significantly’ than the β-TCP group. Please revise the manuscript.

Response 4: Thank you for the suggestion. The above-mentioned changes have been made in the main manuscript under Results section LepR Expression subsection in Lines 461-462. 

Comments 5: It is recommended that a general or detailed explanation of electrospinning technology be added to the introduction rather than a discussion.

Response 5: Thank you for the suggestion. As mentioned, the paragraph on electrospinning has been shifted to the Introduction section from Line 91.

Comments 6: Please present if there is bone regeneration data analyzed between 6 and 12 weeks. The author's in vivo experiment does not seem to agree with the results of Osada et al.

Response 6: Thank you for the comment. We confirm that we did not analyze any specimen between 6 and 12 weeks. Osada et al., evaluated their specimens at two time points, namely 6 weeks and 12 weeks as mentioned. The reference of Osada et al., was added to highlight the fact that the electrospun nanobiomaterial showed active recruitment of osteogenic cells and bone formation at week 12, which is similar to our finding at week 12. Although we noted a significant decline in new bone regeneration by week 12, it has been noted in literature that β-TCP/PLLA/PGA nanobiomaterial can positively enhance osteogenic activity at late stages. The reference of Osada et al., has been retained in our main manuscript file, but the explanation in Discussion section Bone Regenerative capability of β-TCP/PLLA/PGA in comparison to β-TCP subsection from Line 547 has been modified for better understanding.

Comments 7: Please combine figure 16 into one figure. Also, please change the first figure to an image that highlights the regenerative ability of the β-TCP/PLLA/PGA group.

Response 7: We thank you for the suggestion. As mentioned, Figure 16 in Page 23 has been combined into one Figure image. β-TCP is still considered in maxillofacial defects as an ideal bone substitute. We would like to request the reviewer to kindly let us retain the mechanism highlighting the process of bone regeneration in both β-TCP/PLLA/PGA and β-TCP groups, since both biomaterials differ in degradation; bone regeneration associated with both these biomaterials are different. If the reviewer is of the opinion that β-TCP inclusion is unnecessary, then we request the section editor to do the needful.  

Comments 8: Please add a description of the black block in figure 16 or remove the black block.

Response 8: We thank you for the keen observation and comment. The black box in Figure 16 of the β-TCP/PLLA/PGA group diagram was added to represent the enlarged mechanism presented within the subsequent diagram. The black box was modified appropriately to highlight the same.

Comments 9: Please add the sham data in all of the graph and compare with β-TCP and β-TCP/PLLA/PGA group.

Response 9: Thank you for the above suggestion. We apologize for the fact that we cannot perform analyses on the Sham group specimens due to the following reasons: (1) the sham group was created as negative control to enunciate that the critical size defect (4 mm) cannot heal without intervention, (2) only one rat was allotted per time point to the sham group, and (3) inadequate number of animals in sham group does not allow for statistical testing.

Reviewer 2 Report (New Reviewer)

Comments and Suggestions for Authors

The manuscript, titled “Evaluation of bone regenerative capacity of the novel electro-spun nanobiomaterial:β-Tricalcium Phosphate Polylactic acid-co-glycolide (β-TCP/PLLA/PGA) for application in Maxillofacial bone defects: An in-vivo animal study”, undertook a comparative in-vivo investigation into bone regeneration. However, the manuscript's current content lacks novelty and originality, which raises concerns about its suitability for acceptance in its present form. 

  1. The title is too long and needs concise restructuring. 

  1. The 3D reconstructed micro-CT images should be displayed. 

  1. The observation of declined new bone formation at week 12 raises concern, as it suggests potential re-absorption of the formed bone tissue. This requires clarification and further investigation. 

  1. To reinforce the findings presented in Figure 9, inclusion of additional micro-CT images or complementary visual data would significantly support and strengthen the conclusions drawn.

Comments on the Quality of English Language

Moderate editing of English language required

Author Response

Comments 1: The title is too long and needs concise restructuring. 

Response 1: We thank you for the comment. We have shortened and modified the title as follows:

In-vivo evaluation of bone regenerative capacity of the novel nanobiomaterial: β-Tricalcium Phosphate Polylactic acid-co-glycolide (β-TCP/PLLA/PGA) for use in Maxillofacial bone defects.

The above title has also been modified in Page 1 - Title section of the main manuscript.

Comments 2: The 3D reconstructed micro-CT images should be displayed. 

Response 2: Thank you for the comment. The 3-Dimensional (3D) reconstruction of the Micro-CT images for β-TCP/PLLA/PGA, β-TCP, and Sham groups have been displayed as Figure 7 in the main manuscript in Page 10 of the main manuscript. The description for Figure 7 in Line 354 has been changed to clarify the same.

Comments 3: The observation of declined new bone formation at week 12 raises concern, as it suggests potential re-absorption of the formed bone tissue. This requires clarification and further investigation. 

Response 3: We thank you for the comment. It has been evidenced that bone regeneration declines in later stages owing to saturation of the regeneration process. PLLA/PGA degradation does cause accumulation of acidic inflammatory byproducts, those which are completely eliminated from the body naturally. This acidic environment, however, is neutralized by the addition of β-TCP. In-vivo degradation of pure β-TCP takes a longer time. PLLA/PGA on the other hand, degrades much faster in the in-vivo setting. It has been shown that biodegradable materials combined with PLLA/PGA as a result of its faster degradation, decouple and get released in the vicinity quickly. Due to increased amount of β-TCP present in the milieu, bone regeneration at initial stages is comparable between both the materials. We hypothesize that the decline in bone regeneration seen in later stages (week 12) is mostly due to the saturation process and possibly because of lesser β-TCP content in the β-TCP/PLLA/PGA nanobiomaterial. The above explanation has been added to the Discussion section from Line 593.

Comments 4: To reinforce the findings presented in Figure 9, inclusion of additional micro-CT images or complementary visual data would significantly support and strengthen the conclusions drawn.

Response 4: Thank you for the suggestion. We have included a new division – Figure 9A which contains representative Micro-CT images that states the procedure of the Inner defect diameter analysis for better clarity and understanding in Page 12 of the main manuscript.

Reviewer 3 Report (New Reviewer)

Comments and Suggestions for Authors

Subject: Review of Manuscript Submission Nanomaterials_ 2772169

I have carefully reviewed the manuscript titled " Evaluation of bone regenerative capacity of the novel electrospun nanobiomaterial: β-Tricalcium Phosphate Polylactic acid-co-glycolide (β-TCP/PLLA/PGA) for application in Maxillofacial bone defects: An in-vivo animal study" submitted to Nanomaterials for consideration. This research advances the study of the electrospun β-TCP/PLLA/PGA, an ideal nanobiomaterial for inducing bone regeneration by osteoconductivity and bioresorbability in bony defects of maxillofacial region. I think this article is meaningful. It can be accepted after major revision.

1.       The study presents an interesting comparison between β-TCP/PLLA/PGA and pure β-TCP in terms of bone regeneration and bioactivity. However, further clarification is needed on how the different proportions of β-TCP in β-TCP/PLLA/PGA contribute to its bioactivity and osteoconductivity. Does the decreased percentage of β-TCP in the composite material affect its structural integrity or biocompatibility?

2.       While the conclusion highlights the osteoinductive and osteoconductive properties of β-TCP/PLLA/PGA, further elaboration on the underlying mechanisms would strengthen the manuscript. Specifically, how do the physical and chemical properties of the nanobiomaterial contribute to these effects, and what role does the novel fiber shape play in enhancing periostin expression?

3.       The manuscript would benefit from a discussion on the long-term effects and degradation behavior of β-TCP/PLLA/PGA in the maxillofacial bone defect environment. How does the material's degradation align with the rate of new bone formation, and are there any potential adverse reactions or inflammation associated with this process?

4.       The study assesses bioactivity by scoring levels of specific biomarkers. However, it's crucial to understand the temporal dynamics of these biomarkers. How do the levels of Runx2, LepR, Osteocalcin, and Periostin change over time, and what are the implications of these changes for bone regeneration and the maturation of the regenerated bone tissue?

Author Response

Comments 1: The study presents an interesting comparison between β-TCP/PLLA/PGA and pure β-TCP in terms of bone regeneration and bioactivity. However, further clarification is needed on how the different proportions of β-TCP in β-TCP/PLLA/PGA contribute to its bioactivity and osteoconductivity. Does the decreased percentage of β-TCP in the composite material affect its structural integrity or biocompatibility?

Response 1: We thank you for the comment. The material tested is commercially available in the market under the names – ReboSorb®, ReBOSSIS-J®, and ReBOSSIS-MT® for clinical use in human patients. Extensive in-vitro testing has been performed to rule out the possibility of cytotoxicity and also the potentiality to attract and allow osteogenic cell proliferation. This material has also been FDA approved for clinical use since the year 2014. The in-vitro effects are of the commercially available material are well established and the composition is standardized. Many other specialties have conducted in-vivo research analysis on the behavior of the material in various defect models across the world. β-TCP has been combined with other biodegradable materials; but we chose to analyze the in-vivo effects of β-TCP with PLLA/PGA, as the latter is completely degraded from the body with minimal to no adverse effects. PLLA/PGA also does not possess bioactive/osteoconductivity and this feature is applicable to β-TCP alone. Faster degradation of PLLA/PGA releases β-TCP in the milieu, which then brings forth the new bone regeneration. Pure β-TCP takes a longer time to degrade in-vivo. We would also like to state that even at lower concentrations, β-TCP/PLLA/PGA nanobiomaterial demonstrates excellent biocompatibility. The tensile strength of the β-TCP/PLLA/PGA material however may be reduced in overall with increased β-TCP addition, as pure β-TCP material is very brittle with a low tensile strength. We have added detailed in-vitro characteristics of β-TCP/PLLA/PGA nanobiomaterial in the Introduction section from Line 131 for further enunciation of material properties.

Comments 2: While the conclusion highlights the osteoinductive and osteoconductive properties of β-TCP/PLLA/PGA, further elaboration on the underlying mechanisms would strengthen the manuscript. Specifically, how do the physical and chemical properties of the nanobiomaterial contribute to these effects, and what role does the novel fiber shape play in enhancing periostin expression?

Response 2: Thank you for the query. The properties of the nanobiomaterial that contribute to the osteoinductive and osteoconductive effects have been put forth in the Discussion section from Line 572. The mechanism by which β-TCP/PLLA/PGA nanobiomaterial induces bone regeneration is further elaborated in Figure 16, the text in the Discussion section from Line 612 has been highlighted. Electrospinning can generate fibers in the range of 100nm to 5um diameter that mimics the shape of Extracellular matrix (ECM). The orderly shape of the electrospun fibers enhances level of tissue maturation by highly ordered ECM and higher periostin expression. The precise molecular mechanisms underlying the same are unclear and further research is needed to provide conclusive results. This explanation has been added to the Discussion section from Line 677.

Comments 3: The manuscript would benefit from a discussion on the long-term effects and degradation behavior of β-TCP/PLLA/PGA in the maxillofacial bone defect environment. How does the material's degradation align with the rate of new bone formation, and are there any potential adverse reactions or inflammation associated with this process?

Response 3: We thank you for the question posed. It has been well remarked in literature that PLLA/PGA possesses good biocompatibility, mouldability, processability, and control over degradation time by varying compositions of the component materials. PLLA/PGA does not possess any adverse degradation effects as observed with PLLA or PDLA.Though the PLLA in PLLA/PGA can result in production of acidic byproducts during degradation, this environment is neutralized by β-TCP. PLLA/PGA is non-toxic and is completely excreted from the body naturally. β-TCP has a slower degradation than PLLA/PGA, being nearly 2.5 years in vivo. The degraded portions are eventually filled with new bone. It has been noticed that electrospun fibers containing β-TCP and PLGA degraded faster than membranes owing to surface roughness and wettability. We did not perform long-term degradation assessment in the present study; this limitation will be corrected in our future research. The cotton-like structure of our material allows faster water absorption thereby faster degradation, elimination, and bone formation. It is also vital to be noted that we did not observe any inflammatory cells/ giant cells in the vicinity of both the biomaterials in initial as well as late phases, suggestive of good biocompatibility. The explanation concerning material degradation has been added to the Discussion section from Line 593.

Comments 4: The study assesses bioactivity by scoring levels of specific biomarkers. However, it's crucial to understand the temporal dynamics of these biomarkers. How do the levels of Runx2, LepR, Osteocalcin, and Periostin change over time, and what are the implications of these changes for bone regeneration and the maturation of the regenerated bone tissue?

Response 4: Thank you for the above query. More details on biomarker expression and its relevance in bone regeneration has been added in the Discussion section under Significance of Biomarker Expression during bone regeneration subsection in Lines 624, 628, 630, and 646. We agree with the reviewer’s comment on incorporating additional data regarding the biomarkers and have highlighted explanation of biomarker relevance into the Discussion section.

Round 2

Reviewer 2 Report (New Reviewer)

Comments and Suggestions for Authors

1. The depiction in Figure 7 provides a slice or top view of the 3D reconstructed micro-CT images. Please provide micro-ct images in a way that comprehensively represents the volumetric aspect of the regenerated bone tissue.

2. There are too many figures in this manuscript, please reduce the number of figures by categorizing multiple images into single figures for enhanced clarity. For example, merging figures 2, 3, and 4 could illustrate the experiment process more clearly, while combining figures 7, 8, 9, and 10 can strengthen the bone regeneration performance of materials. Additionally, figures 5, and 6 should be put into the supplementary files. 

Comments on the Quality of English Language

Minor editing of English language required

Author Response

Comments 1: The depiction in Figure 7 provides a slice or top view of the 3D reconstructed micro-CT images. Please provide micro-ct images in a way that comprehensively represents the volumetric aspect of the regenerated bone tissue.

Response 1: We thank you for the suggestion. b-TCP bone substitute is accredited owing to its quality of mimicking the natural bone morphology; we have mentioned this in Line 56 of Introduction. Though it is possible to differentiate other biodegradable materials with different compositions in volumetric reconstruction due to their densities, differentiation of newly formed woven bone and bone particles/ bone nodule identification relative to scaffolds fabricated with b-TCP may not be discernible on a macroscopic level. We have added two more views of the 3D model in addition to the Buccal view, namely Lingual view and Sagittal view. We sincerely hope that these extra images aid better understanding.

Comments 2: There are too many figures in this manuscript, please reduce the number of figures by categorizing multiple images into single figures for enhanced clarity. For example, merging figures 2, 3, and 4 could illustrate the experiment process more clearly, while combining figures 7, 8, 9, and 10 can strengthen the bone regeneration performance of materials. Additionally, figures 5, and 6 should be put into the supplementary files. 

Response 2: Thank you for the comment. As per the reviewer’s suggestion, Figures 2, 3, and 4 have been combined into one single Figure. Figures 5 and 6 – changed to Figures 3 and 4 respectively have been removed from the main manuscript and added into supplementary files. Figures 7, 8, 9, and 10 have been combined into one Figure – Figure 3. The total number of Figures has been reduced from 16 to 9, inclusive of two supplementary files.

Reviewer 3 Report (New Reviewer)

Comments and Suggestions for Authors

I am satisfied with the revision. It can be accepted in its present form.

Author Response

Dear Reviewer,

We offer our heartfelt thanks for your kind comment in aiding the accepting of our research paper. We also would like to thank you for your valuable time in reviewing and refining the contents of our research article. We will include all the comments and correct any shortcomings suggested in our future research endeavors.

Round 3

Reviewer 2 Report (New Reviewer)

Comments and Suggestions for Authors

No comments

This manuscript is a resubmission of an earlier submission. The following is a list of the peer review reports and author responses from that submission.

Round 1

Reviewer 1 Report

Comments and Suggestions for Authors

This manuscript describes the in vivo comparison and evaluation of b-TCP/PLLA/PGA scaffold for bone regeneration. The authors fabricated an electrospun fiber made of β-TCP and PLLA/PGA, which are bioabsorbable materials, to heal maxillofacial bone defects. Additionally, its effectiveness was compared with that of the β-TCP single scaffold in an in vivo environment. The results showed that the b-TCP/PLLA/PGA group had the highest cytocompatibility and bone regeneration properties. Overall, this study presents an approach for bone regeneration using a biocompatible b-TCP/PLLA/PGA integrated electrospun fiber and reports the effectiveness of the novel electrospun nanomaterials through comparisons between several biomarkers: however, the reviewer thinks that this study needed to show the more exact in vivo results and in vitro supplement study to prove the effect of b-TCP/PLLA/PGA on bone regeneration. Therefore, the reviewer thinks that this manuscript needed to modify for publication of ‘namaterials’. Below are some comments to improve the quality of this work:

Specific comments

1) Please evaluate the cytotoxicity and cell affinity of the synthesized b-TCP/PLLA/PGA electrofiber in vitro.

2) Please describe the advantages of b-TCP/PLLA/PGA electrofiber compared to the b-TCP/PLLA group in addition to the b-TCP group in more detail in the introduction.

3) Please indicate the scale bar in Figure 7.

4) Please briefly describe in the introduction the cases in which electrospun nanobiomaterials and biomaterials have been applied to regenerative therapy.

5) In Figures 7 and 8, please describe whether defect areas were given to all experimental groups the same. Also, please describe the concentration selection and quantification criteria for each group. (b-TCP: 16 mg, b-TCP/PLLA/PGA: 3 mg)

6) In Figure 9, the bone regeneration rate of the b-TCP group is more than twice that of the b-TCP/PLLA/PGA group, which is not an insignificant difference.

7) To indicate statistical significance, please describe and use a more precise unit. (e.g. *, **, ***)

8) In H&E Staining in Figure 11, please express defective areas and regenerated areas in an easy-to-see manner. (e.g. arrows)

9) Please integrate the IHC analysis images and IHC OD score graphs for the four biomarkers Runx2, OCN, LepR, and periostin.

10) Please present cleary the possible mechanism of bone regeneration using  b-TCP/PLLA/PGA in Figure 16.

11) Please evaluate the level of expression of angiogenic factors, such as VEGF,

during b-TCP/PLLA/PGA treatment in vivo.

Reviewer 2 Report

Comments and Suggestions for Authors

This in vivo study demonstrates the bone regeneration capacity of the electrospun biomaterial-based scaffolds. The experimental groups consist of β-TCP and the combination of β-TCP/PLLA/PGA to investigate the degree of bone regeneration up to 12 weeks. As the authors mention, electrospinning is one of the most popular methods due to its easy handling and reproducibility. Also, the proposed nanomaterials are well-known for their biocompatibility. Regarding that, this manuscript is in the right direction. However, there are some concerns that should be addressed before publication as follows.

1.     Do not omit abbreviation process although certain terms are conventionally used. (Ex, PLLA, PGA, PBS, and others). Please abbreviate when they first appear.

2.     In Figure 1, it is hard to see the scale bars in some images.

3.     In Figure 4(f), it would be better it there was a magnified image of surgical region.

4.     In Figure 6, some letters are too small so that hard to read.

5.     Page 18, line 458, there is a capitalization error.

6.     Figure 8, is there any reasonable explanation BV/TV ratio has dropped after week 4? Maybe because of the saturated bone regeneration process?

7.     Page 19, line 500, the font is not unform (non-italic).

8.     Figure 16 should be revised. Please use a software tool to draw a figure. Hand-drawn images are not generally permitted in any scientific journals.

9.     The reviewer is in an opinion of that one of the innovations the authors aim to emphasize through this manuscript is the convenient fabrication method, electrospinning. However, there is a very small portion dealing with it. Therefore, it would be worth mentioning what’s good about using electrospinning to perform this study.

10.  The first paragraphs of each section in the discussion (Section 4) consist of too much background knowledge, which seem to be more suitable for introduction. It is recommended to shorten them and focus more on the perspective of discussion.

11.  Adding a short paragraph (or a section) of limitation would enhance the quality of this manuscript.